# 24 hour consultant obstetrician presence on the labour ward and intrapartum outcomes in a large unit in England: A time series analysis

Sharon Morad[1]*, David Pitches[2], Alan Girling[3], Beck Taylor[3], Vikki Fradd[4], Christine MacArthur[3], Sara Kenyon[3]

**1** Birmingham Women's Hospital, Birmingham, United Kingdom, **2** Directorate of Public Health and Wellbeing, Dudley Metropolitan Borough Council, Dudley, United Kingdom, **3** Institute of Applied Health Research, University of Birmingham, Birmingham, United Kingdom, **4** Neonatal Unit, Birmingham Heartlands Hospital, University Hospitals Birmingham, Birmingham, United Kingdom

* sharon.morad@nhs.net

**Data Availability Statement:** There are legal and ethical restrictions on sharing the data set. The data is routine English National Health Service

## Abstract

### Objectives

To explore the effect of introducing 24/7 resident labour ward consultant presence on neonatal and maternal outcomes in a large obstetric unit in England.

### Design

Retrospective time sequence analysis of routinely collected data.

### Setting

Obstetric unit of large teaching hospital in England.

### Participants

Women and babies delivered between1 July 2011 and 30 June 2017. Births <24 weeks gestation or by planned caesarean section were excluded.

### Main outcome measures

The primary composite outcome comprised intrapartum stillbirth, neonatal death, babies requiring therapeutic hypothermia, or admission to neonatal intensive care within three hours of birth. Secondary outcomes included markers of neonatal and maternal morbidity. Planned subgroup analyses investigated gestation (<34 weeks; 34–36 weeks; ≥37 weeks) and time of day.

### Results

17324 babies delivered before and 16110 after 24/7 consultant presence. The prevalence of the primary outcome increased by 0.65%, from 2.07% (359/17324) before 24/7 consultant presence to 2.72% (438/16110, P < 0.001) after 24/7 consultant presence which was consistent with an upward trend over time already well established before 24/7 consultant

patient data, and contains potentially identifiable or sensitive patient information. A data sharing agreement is in place between the NHS trust and the University of Birmingham to enable analysis of data, but does not permit sharing beyond the research team. Ethical approval provided by the University of Birmingham Research Ethics Committee, reference ERN_18-0660, contact s.l. cottam@bham.ac.uk. The ethical approval was requested for access to data by the research team only, and not by any external parties. Requests for ethical approval and data access can be addressed to research governance office of the University of Birmingham (researchgovernance@contacts. bham.ac.uk).

**Funding:** CMcA, SK, AG and BT were funded by the National Institute for Health Research (NIHR) Collaborations for Leadership in Applied Health Research and Care (CLAHRC) West Midlands, UK and Applied Research Collaboration, https:// warwick.ac.uk/fac/sci/med/about/centres/arc-wm/. The views expressed are those of the authors and not necessarily those of the NIHR or the UK Department of Health and Social Care. The funder had no role in study design, data collection and analysis, decision to publish, or preparation of the manuscript.

**Competing interests:** Sharon Morad reports that she is married to one of the co-authors (DP). David Pitches reports that he is married to the corresponding author (SM). Alan Girling has nothing to disclose. Vikki Fradd has nothing to disclose. Beck Taylor reports funding from the NIHR during the study duration. Christine MacArthur reports funding from the NIHR during the study duration. Sara Kenyon reports funding from the NIHR during the study duration. Our competing interests statement does not alter our adherence to PLOS ONE policies on sharing data and materials.

presence began (OR 1.09 p.a.; CI 1.04 to 1.13). Overall, there was no change in this trend associated with the transition to 24/7. However, in babies born ≥37 weeks gestation, the upward trend was reversed after implementation of 24/7 (OR 0.67 p.a.; CI 0.49 to 0.93; P = 0.017). No substantial differences were shown in other outcomes or subgroups.

## Conclusions

Overall, resident consultant obstetrician presence 24/7 on labour ward was not associated with a change in a pre-existing trend of increasing adverse infant outcomes. However, 24/7 presence was associated with a reversal in increasing adverse outcomes for term babies.

## Introduction

The UK triennial national enquiries into maternal and perinatal deaths from the 1990s and early 2000s all highlighted suboptimal care associated with intra-partum fetal deaths and maternal deaths; and the late input of an obstetric consultant was cited as a contributory factor to these outcomes [1, 2]. National Patient Safety Agency data suggested that adverse incidents were more likely to occur at night when a consultant was unlikely to be present [3]; and an association between adverse perinatal outcomes, including mortality, and births at night was shown in a seven year registry-based cohort study including all hospitals in the Netherlands [4, 5]. In England a large observational study based on routine Hospital Episode Statistics data for all births over two years (n = 1332835) showed that perinatal mortality was worse than for births at weekends than on weekdays [6]. In contrast, studies of units in the United States where there was 24/7 specialist obstetric presence did not detect associations between outcomes and time of birth [7, 8].

In 2005 the UK Royal College of Obstetricians and Gynaecologists (RCOG) recommended that by 2010 all maternity units with >5000 births per year should have resident obstetric consultant presence 168 hours per week (i.e. 24/7 consultant presence) [9]. It was theorised that increasing the presence of obstetric consultants on the labour ward would increase the timely recognition of problems and instigate appropriate action more rapidly, whilst avoiding unnecessary intervention. Greater experience should also result in better technical and non-technical skills, and the presence of an additional senior obstetrician could provide the ability to manage multiple time critical events simultaneously. The result should be improved patient safety, reduced litigation, and better staff morale and teamwork [10, 11]. It was recognised that this was challenging to implement, and the RCOG made interim recommendations to improve safety and training where 24/7 cover was not in place [12]. Systematic reviews of UK studies examining the relationship between consultant presence and adverse events however have failed to demonstrate any reduction in adverse outcomes with increasing consultant presence, though included studies were small and quality was low [13, 14]. In 2016 and 2017 the UK RCOG changed their recommendations, suggesting that alternatives to 24/7 consultant presence should be explored [15, 16].

A large National Health Service Trust in the UK decided to implement 24 hour consultant cover in one of its two maternity units; by 2014 this Trust became one of only two hospitals in the UK to have fully implemented the 2005 RCOG recommendations with 168 hours of resident consultant cover on the labour ward, in addition to the three tiers of non-consultant doctors. Given that more evidence is needed to determine whether 24/7 consultant presence on labour ward could reduce adverse neonatal and maternal outcomes, the aim of this study was

to explore the effect of introducing 24/7 consultant presence using a time-series approach over a six year period. The research team analysed routine health service data to establish the effect of increased consultant presence on key outcomes.

## Methods

### Objectives

The primary objective was to explore the effect of introduction of 24/7 labour ward obstetric consultant presence on pre-specified neonatal and maternal outcomes over a six year period 2011–2017. Additional objectives were to explore effects of 24/7 consultant presence within gestation specific sub-groups, and for day- and night-time births.

### Setting

The NHS Trust comprises two hospitals with obstetric units; the study was conducted in the larger unit (Unit A) with approximately 6000 births annually, and level 3 neonatal care unit (defined by the British Association of Perinatal Medicine [17]). The other unit (Unit B) has approximately 3600 births annually and level 1 neonatal unit.

### Intervention

Throughout the study period labour ward was staffed with 3 non-consultant grade obstetricians 24/7. Prior to July 2014 consultants were only physically present on labour ward from 08:00–20:00, and on-call from home overnight. Unit A commenced full 24/7 resident consultant presence (i.e. 168 hours/week) in July 2014. Existing consultants could choose whether to participate in the resident night cover and five additional consultants were appointed (cost £778,000 per year) with resident night cover part of their job plan. The resident consultant obstetrician night cover was in addition to existing staff on the Unit, i.e. not a replacement/ substitution of other team members. Consultant duties, regardless of time of day, included running the labour ward, overseeing ward rounds, supervising the decision-making and procedures of non-consultant obstetricians, and carrying out higher risk deliveries when required. After three years the hospital ceased the intervention in 2017 due to funding constraints (before the evaluation presented in this paper was undertaken).

### Subjects

Pseudonymised retrospective data for women and babies in Unit A from 1 July 2011 to 30 June 2017 –three years before and three years after introduction of the intervention, up to the point at which it was discontinued and 24/7 obstetric consultant cover was withdrawn by the hospital. Births before 24 weeks gestation were not included in line with UK national definitions of viability [18], nor births by planned caesarean section since consultants were always responsible for these births, on elective theatre lists 'in hours'. In 2014 a 'Preterm Pathway' was introduced which directed women to attend unit A directly if presenting before 30 weeks with a singleton pregnancy or 34 weeks with twins, as Unit B is a Special Care Unit and provides only stabilisation and short term High Dependency or Intensive Care. The Preterm Pathway did not involve any change to obstetric staffing levels in Unit A.

### Outcome measures

The primary outcome was a composite neonatal outcome which comprised intrapartum stillbirth, neonatal death with and without congenital abnormality, babies requiring therapeutic

hypothermia and/or admission to Neonatal Intensive Care Unit (NICU) within three hours of birth.

The items within the composite were chosen by a multi-professional team comprising obstetricians, midwives, neonatologists, and researchers. Items were included as serious outcomes that could plausibly be affected by presence of a consultant obstetrician; and that outcome definitions and data collection were consistent throughout study period.

Secondary neonatal outcomes included individual components of the primary composite plus 5 minutes Apgar under 7, babies that required ventilation (any mechanical respiratory support via an endotracheal tube), babies with seizures within the first 28 days, perinatal mortality and its separate components ((stillbirth (antenatal and intrapartum), early neonatal death (before 7 completed days of life), and late neonatal death (after 7 completed days but before 28 days after birth)).

Secondary maternal outcomes drawn from the literature were selected according to relevance and data availability. They included mode of birth (spontaneous vaginal, instrumental, unplanned caesarean section categories 1, 2, and 3 defined by NICE [19]), postpartum haemorrhage ≥1000ml defined by national guidance [20], and use of Fresh Frozen Plasma (FFP) or cryoprecipitate. In maternal outcomes use of FFP or cryoprecipitate was chosen instead of use of packed red cells as protocols for use of packed red cells changed over the study period. Maternal deaths were considered but not included due to their rarity.

### Analysis

Data were extracted from routinely maintained electronic maternity information systems, not created specifically for study purposes. The maternity information system changed partway through the study period, but the same neonatal system was used throughout. There was a decrease in completion of some data fields for a few months immediately following introduction of the new system however maternal haemorrhage was the only field to be affected. The data-set was pseudonymised and summarised by intervention status (pre- and post-24/7 consultant presence). Computations were carried out in STATA15 [21].

Logistic regression was used to investigate the impact of 24/7 consultant presence, supplemented by Interrupted Time Series (ITS) analysis. Two alternative time-series analyses were considered. Both entailed fitting separate time-trends before and after the introduction of 24/7 cover. In the first analysis the linear trends were fitted to the series of monthly averaged outcome rates with autoregressive error-terms. In practice the autocorrelations were close to zero. Results in the paper derive from an alternative analysis in which segmented logistic regression models were fitted to individual outcome data with a cut-point at the introduction of 24/7 cover. This ignores autocorrelations in the outcome series, but results are effectively indistinguishable from the autoregressive model. Sub-group analyses were conducted by weeks of gestation (under 34 weeks; 34–36 weeks; 37 weeks and over) and time of day (8:00–19:59 and 20:00–08:59, chosen to match 'Day' and 'Night' shift hours of the resident consultant). P-values are derived from two-tailed Z-tests with P = 0.05 as the threshold of significance throughout.

Assuming 6500 births per year—19500 before and 19500 after implementation—the study had 80% power (2-sided test of proportions, 5% significance) to detect a fall in the incidence of the composite outcome from 7.1 to 4.9 per 1000 births. The baseline incidence of 7.1 (per 1000 births) is based on 2015 data from the unit and comprises: 2.06 neonatal deaths (per 1000); 0.54 intrapartum stillbirths (per 1000) estimated as 10% of total stillbirths [15]; 1.50 cases of therapeutic cooling (per 1000) [16]; and an assumed rate of NICU admission of 3.0 babies per 1000. The power was calculated before study-data were available. In the event the assumed NICU admission rate proved to be a considerable under-estimate. (Post-hoc power calculation

assuming a more realistic baseline NICU admission-rate of 20 babies per 1000 would give 90% power to detect a fall in the composite outcome from 24.1 to 19.3 per 1000 births.)

### National survey of practice

To provide national context for the local study of consultant presence, a survey of clinical directors of all maternity units in the UK was conducted in July 2019 to explore their current consultant staffing levels on labour ward (reported as hours per week of consultant presence). Enquiry was also made into what clinical directors thought were main benefits of 24/7 consultant presence and barriers to implementation. A letter and paper survey was sent by post to clinical directors of every maternity unit in the UK (n = 196) in July 2019, followed by reminder letters at one and two months. Participants were able to respond by post, email, or via an online survey platform www.onlinesurveys.ac.uk [22]. The survey included quantitative and quantitative questions (S1 File). Microsoft Excel was used for analysis. Descriptive summary statistics (median, interquartile range) were calculated for quantitative data, and qualitative free text responses were analysed thematically [23]. The survey was undertaken concurrently with the main study: its findings did not influence the study methodology.

### Ethical approval

Ethical review was obtained from the University of Birmingham (ERN_18–0660), including approval to use anonymous retrospective routinely gathered NHS patient data without consent, as researchers did not have access to patient records, and were not able to track or link individuals in the dataset.

## Results

There were 33,434 babies born (33,051 women who gave birth) over six years; this comprised 17,324 babies (17,131 women) before and 16,110 babies (15,920 women) after 24/7 consultant presence.

### Demographics

Demographic characteristics and birth-risk categorisation are in Table 1. The number of births declined somewhat over the study period (from 5,775 per annum to 5,370 per annum). In both time periods 54% of births occurred at night. The proportion of births of <34 weeks' gestation increased from 2.82% to 3.23%, likely due to introduction of the Preterm Pathway in 2014, which drew additional premature births into unit A from unit B.

### Primary outcome

The proportion of babies experiencing the primary (composite) outcome increased from 2.07% (359/17324) to 2.72% (438/16110) after 24/7 labour ward consultant presence was introduced in July 2014 (OR 1.32; CI 1.15 to 1.52; P < 0.001. Table 2). The increase was mainly due to an increased proportion of babies transferred to NICU within three hours of birth (1.86% (322/17324) rising to 2.51% (405/16110); OR 1.36; CI 1.17 to 1.58. Table 2). NICU admission dominated the composite outcome and was present in 91% of all its occurrences. The difference in proportions is consistent with a general upward trend in the prevalence of the composite outcome over time, already well established before July 2014 (OR 1.09 p.a.; CI 1.04 to 1.13. Table 2). The upward trend continued after July 2014, but without any significant change associated with transition to 24/7 presence (OR 0.93 pa; CI 0.79 to 1.09; P = 0.378. Table 2 and Fig 1a).

**Table 1. Maternal demographics\*.**

| | | Before 24/7 | After 24/7 |
|---|---|---|---|
| | | (N = 17,131) | (N = 15,920) |
| Age: | median (Q1, Q3) | 28 (24, 32) | 29 (25, 33) |
| | N = | 17,131 | 15,920 |
| BMI: | median (Q1, Q3) | 25 (22, 29) | 25 (22, 29) |
| | N (available) = | 14,437 | 11,113 |
| Ethnic Origin (No., %) | | | |
| | African | 859 (5.01) | 813 (5.11) |
| | Asian | 6,365 (37.15) | 5,893 (37.02) |
| | European | 7,331 (42.79) | 6,648 (41.76) |
| | Other | 825 (4.82) | 1,055 (6.63) |
| | Mixed | 306 (1.79) | 292 (1.83) |
| | Not Stated | 1,445 (8.44) | 1,219 (7.66) |
| | Total | 17,131 (100.0) | 15,920 (100.0) |
| Index of Multiple Deprivation Quintile (No., %) | | | |
| | 1 | 12,303 (71.93) | 11,434 (71.90) |
| | 2 | 1,854 (10.84) | 1,695 (10.66) |
| | 3 | 1,222 (7.14) | 1,211 (7.62) |
| | 4 | 840 (4.91) | 781 (4.91) |
| | 5 | 886 (5.18) | 781 (4.91) |
| | Total (available) | 17,105 (100.0) | 15,902 (100.0) |
| Risk-Category (No., %)\*\* | | | |
| | Low | 6,734 (40.55) | 5,892 (39.72) |
| | High | 9,871 (59.45) | 8,942 (60.28) |
| | Total (available) | 16,605 (100.0) | 14,834 (100.0) |
| Gestation (No., %) | | | |
| | 24+0 to 33+6 wks | 483 (2.82) | 514 (3.23) |
| | 34+0 to 36+6 wks | 952 (5.56) | 968 (6.08) |
| | 37 wks or greater | 15,677 (91.51) | 14,435 (90.67) |
| | Unknown | 19 (0.11) | 3 (0.02) |
| | Total | 17,131 (100.0) | 15,920 (100.0) |
| Hour of Birth (No., %) | | | |
| | 08:00–19:59 ("Day") | 7942 (46.36) | 7321 (45.99) |
| | 20:00–07:59 ("Night") | 9189 (53.64) | 8599 (54.01) |
| | Total | 17,131 (100.0) | 15,920 (100.0) |

\* Parity was not collected electronically so could not be included.

\*\* Maternal risk category was taken from that assigned at booking. The definition of low and high risk women did not change over the study period. 'Low risk' included those women on the standard NHS tariff maternity pathway for payment, and 'high risk' included those on the intermediate and intensive pathway.

## Secondary outcomes

**Neonatal outcomes.**   Overall increases were seen in admission to NICU, ventilation, and emergency caesarean, but these were not associated with the implementation of 24/7 consultant presence. The increased prevalence of ventilation (1.48% (256/17324) to 2.07% (333/16110); OR 1.41; CI 1.19 to 1.66; P < 0.001. Table 2) is explicable in terms of a pre-existing trend (OR 1.12 pa; CI 1.01 to 1.25. Table 2), unaffected by transition to 24/7 consultant presence. Other neonatal outcomes showed no clear differences over time (Table 2).

**Table 2. Primary and secondary outcomes.**

| | Comparison between outcomes before and after July 2014 | | | | Time Series Analysis | | | |
| --- | --- | --- | --- | --- | --- | --- | --- | --- |
| | – i.e. without and with 24/7 consultant cover | | | | Overall slope 2011–2016 | Slope until July 2014 | Change in slope from July 2014 | |
| | N (%) | N (%) | Odds-Ratio | P-value | Odds-Ratio (p.a.) | Odds-Ratio (p.a.) | Odds-Ratio (p.a.) | P-value |
| | Before July 2014 | From July 2014 | (95% CI) | | (95% CI) | (95% CI) | (95% CI) | |
| **Primary Outcomes** | | | | | | | | |
| Numbers of Babies | 17,324 | 16,110 | | | | | | |
| Primary (Composite) Outcome | 359 (2.07) | 438 (2.72) | 1.32 (1.15–1.52) | < 0.001 | 1.09 (1.04–1.13) | 1.13 (1.03–1.24) | 0.93 (0.79–1.09) | 0.378 |
| **Secondary Outcomes (Neonatal)** | | | | | | | | |
| Intrapartum Still Birth | 17 (0.10) | 9 (0.06) | 0.57 (0.25–1.28) | 0.172 | 0.74 (0.58–0.94) | 0.82 (0.53–1.27) | 0.75 (0.27–2.05) | 0.574 |
| Neonatal Death | 50 (0.29) | 61 (0.38) | 1.31 (0.90–1.91) | 0.154 | 1.09 (0.98–1.22) | 0.92 (0.72–1.18) | 1.39 (0.90–2.16) | 0.134 |
| Therapeutic Hypothermia | 22 (0.13) | 25 (0.16) | 1.22 (0.69–2.17) | 0.493 | 1.03 (0.87–1.21) | 1.58 (1.07–2.33) | 0.41 (0.21–0.82) | 0.012 |
| NICU admission within 3 hours | 322 (1.86) | 405 (2.51) | 1.36 (1.17–1.58) | < 0.001 | 1.10 (1.05–1.14) | 1.14 (1.04–1.26) | 0.92 (0.77–1.09) | 0.322 |
| Apgar5 < 7 | [a]256 (1.49) | [b]265 (1.66) | 1.12 (0.94–1.33) | 0.216 | 1.05 (0.77–1.41) | 0.95 (0.85–1.06) | 1.15 (0.94–1.40) | 0.180 |
| Seizures | 49 (0.28) | 34 (0.21) | 0.75 (0.48–1.16) | 0.189 | 0.89 (0.79–1.01) | 0.97 (0.75–1.26) | 0.83 (0.50–1.37) | 0.461 |
| Ventilation required | 256 (1.48) | 333 (2.07) | 1.41 (1.19–1.66) | < 0.001 | 1.09 (1.04–1.14) | 1.12 (1.01–1.25) | 0.95 (0.78–1.14) | 0.573 |
| Still Birth (Ante- or Intra-partum) | 118 (0.68) | 134 (0.83) | 1.22 (0.95–1.57) | 0.112 | 1.04 (0.97–1.12) | 0.91 (0.78–1.07) | 1.29 (0.97–1.73) | 0.080 |
| Early (< 7 days) Neonatal Death | 34 (0.20) | 43 (0.27) | 1.36 (0.87–2.14) | 0.180 | 1.14 (1.00–1.30) | 0.99 (0.73–1.33) | 1.32 (0.78–2.22) | 0.305 |
| Late (7 to 28 days) Neonatal Death | 12 (0.07) | 18 (0.11) | 1.61 (0.78–3.35) | 0.199 | 1.11 (0.90–1.36) | 1.05 (0.65–1.69) | 1.11 (0.48–2.54) | 0.812 |
| **Secondary Outcomes (Maternal)\*** | | | | | | | | |
| Women | 17,111 | 15,917 | | | | | | |
| Spontaneous Vaginal Birth | 12005 (70.16) | 11053 (69.44) | 0.97 (0.92–1.01) | 0.156 | 0.99 (0.97–1.00) | 0.99 (0.96–1.02) | 1.00 (0.94–1.05) | 0.921 |
| Instrumental | 2081 (12.16) | 1859 (11.68) | 0.96 (0.89–1.02) | 0.177 | 0.98 (0.96–1.00) | 1.01 (0.97–1.06) | 0.94 (0.87–1.01) | 0.112 |
| Emergency C-Section | 3025 (17.68) | 3005 (18.88) | 1.08 (1.02–1.15) | 0.005 | 1.03 (1.02–1.05) | 1.01 (0.97–1.05) | 1.05 (0.98–1.12) | 0.165 |
| Postpartum Haemorrhage | [c]717 (4.21) | [d]876 (5.93) | 1.43 (1.30–1.59) | < 0.001 | 1.10 (1.07–1.13) | 1.24 (1.16–1.33) | 0.78 (0.70–0.88) | < 0.001 |
| Blood Products used | 48 (0.28) | 42 (0.26) | 0.94 (0.62–1.42) | 0.772 | 0.95 (0.84–1.07) | 0.89 (0.69–1.15) | 1.14 (0.71–1.85) | 0.583 |

\*23 cases excluded where gestation and/or mode of birth not known (20 before and 3 afterward).

[a-d] Data incomplete for Apgar ([a] N = 17135, [b] N = 15927) and Postpartum Haemorrhage ([c]N = 17034, [d]N = 14775).

**Maternal outcomes.** An overall increase was seen in emergency caesarean, but this was not associated with the implementation of 24/7 consultant presence. The rise in postpartum haemorrhage levelled off after introduction of 24/7 consultant presence. Emergency caesarean sections were more common after July 2014, rising from 17.68% (3025/17111) to 18.88%

Fig 1a: All Neonates

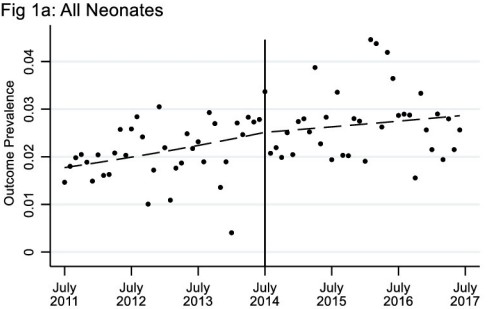

Fig 1b: 37+ weeks gestation

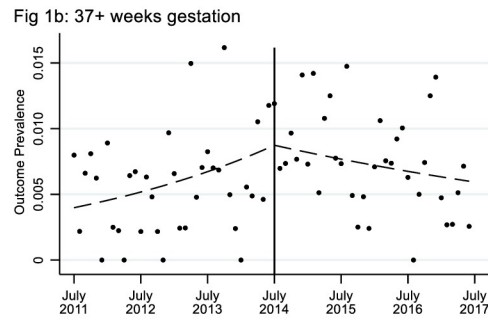

Fig 1c: 34 weeks to 36 weeks + 6 days gestation

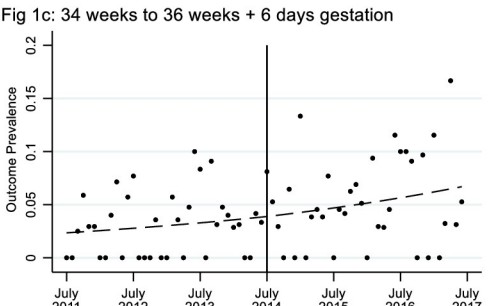

Fig 1d: 24 weeks to 33 weeks + 6 days gestation

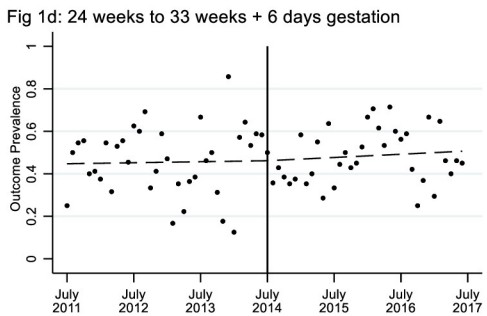

**Fig 1. Composite neonatal outcome by month of birth, and stratified by gestation.** The fitted lines are from (segmented) logistic regressions with a slope-change in July 2014 (i.e. when 24 hour cover was initiated). The increase over time is generally unaffected by 24-hour cover, except for full-term babies (Fig 1b) where the prevalence tends to fall after July 2014.

(3005/15917) of all births (OR 1.08; CI 1.02 to 1.15; P = 0.005. Table 2). The trend analysis is inconclusive here, and does not show a significant change associated with 24/7 presence (Table 2). The prevalence of postpartum haemorrhage increased from 4.21% (717/17034) to 5.93% (876/14775; OR 1.43; CI 1.30 to 1.59; P < 0.001). Here an upward trend was already evident before 24/7 consultant presence (OR 1.24 pa; CI 1.16 to 1.33). There was a substantial decrease in the slope of the line after July 2014 (OR 0.78 pa; CI 0.70 to 0.88; P < 0.001)–consistent with a beneficial impact of 24/7 presence, which nullifies the year-on-year increase from this point onwards (Table 2). The haemorrhage results must be treated with caution however since the proportion of women for whom blood loss was recorded fell from 99.6% before 24/7 consultant presence to only 92.8% after July2014. Thus the recorded rates of haemorrhage are highest when the data quality is low. Other maternal outcomes showed no clear differences over time (Table 2).

## Subgroup analyses

As expected, the prevalence of the adverse outcomes in both subgroup time periods was highest for births < 34 weeks and lowest for term (≥ 37 weeks) births (Tables 3 and 5). For term birth, however, the upward trend in the primary composite outcome was reduced after July 2014 (OR 0.67 pa; CI 0.49 to 0.93; P = 0.017. Table 3 and Fig 1). This included a reduction in the upward trend for term babies admitted to NICU (OR 0.63, CI 0.44 to 0.89; P = 0.009) and term babies requiring therapeutic hypothermia (OR 0.45; CI 0.22 to 0.93; P = 0.030).

Analysis by time of day revealed that adverse outcomes were more prevalent in the daytime (Tables 4 and 5). The trend in babies born at night of any gestation who required

**Table 3. Neonatal outcomes stratified by weeks gestation*.**

| | Comparison between outcomes before and after July 2014 | | | | Time Series Analysis | | | |
| | – i.e. without and with 24/7 consultant cover | | | | Overall slope 2011–2016 | Slope until July 2014 | Change in slope from July 2014 | |
| | N (%) | N (%) | Odds-Ratio | P-value | Odds-Ratio (p.a.) | Odds-Ratio (p.a.) | Odds-Ratio (p.a.) | P-value |
| | Before July 2014 | From July 2014 | (95% CI) | | (95% CI) | (95% CI) | (95% CI) | |
|---|---|---|---|---|---|---|---|---|
| **24 wks to 33 wks + 6 days:** | | | | | | | | |
| Numbers of Babies | 521 | 571 | | | | | | |
| Primary (Composite) Outcome | 239 (45.87) | 274 (47.99) | 1.09 (0.86–1.38) | 0.485 | 1.04 (0.97–1.11) | 1.02 (0.87–1.19) | 1.04 (0.79–1.38) | 0.768 |
| Intrapartum Still Birth | 7 (1.34) | 5 (0.88) | 0.65 (0.20–2.06) | 0.462 | 0.79 (0.56–1.11) | 1.23 (0.63–2.42) | 0.31 (0.07–1.47) | 0.139 |
| Neonatal Death | 32 (6.14) | 33 (5.78) | 0.94 (0.57–1.55) | 0.800 | 1.00 (0.87–1.15) | 0.75 (0.54–1.04) | 1.79 (0.98–3.26) | 0.060 |
| Therapeutic Hypothermia | 2 (0.38) | 0 (0.00) | | | | | | |
| NICU admission within 3 hours | 223 (42.80) | 260 (45.53) | 1.12 (0.88–1.42) | 0.364 | 1.05 (0.98–1.12) | 1.02 (0.87–1.19) | 1.07 (0.81–1.41) | 0.652 |
| **34 wks to 36 wks + 6 days:** | | | | | | | | |
| Numbers of Babies | 1032 | 1042 | | | | | | |
| Primary (Composite) Outcome | 31 (3.00) | 54 (5.18) | 1.76 (1.12–2.77) | 0.013 | 1.20 (1.06–1.37) | 1.19 (0.87–1.62) | 1.02 (0.61–1.71) | 0.929 |
| Intrapartum Still Birth | 3 (0.29) | 1 (0.10) | 0.33 (0.03–3.17) | 0.337 | | | | |
| Neonatal Death | 9 (0.87) | 9 (0.86) | 0.99 (0.39–2.50) | 0.984 | 1.07 (0.82–1.40) | 0.84 (0.46–1.53) | 1.62 (0.55–4.81) | 0.382 |
| Therapeutic Hypothermia | 1 (0.10) | 2 (0.19) | 1.98 (0.18–21.9) | 0.577 | | | | |
| NICU admission within 3 hours | 24 (2.33) | 48 (4.61) | 2.03 (1.23–3.34) | 0.005 | 1.25 (1.08–1.44) | 1.32 (0.93–1.88) | 0.90 (0.51–1.59) | 0.720 |
| **≥ 37 weeks:** | | | | | | | | |
| Numbers of Babies | 15751 | 14494 | | | | | | |
| Primary (Composite) Outcome | 89 (0.56) | 110 (0.76) | 1.35 (1.02–1.78) | 0.037 | 1.07 (0.99–1.16) | 1.30 (1.08–1.57) | 0.67 (0.49–0.93) | 0.017 |
| Intrapartum Still Birth | 7 (0.04) | 3 (0.02) | 0.47 (0.12–1.80) | 0.269 | 0.65 (0.43–1.00) | 0.56 (0.27–1.16) | 1.57 (0.29–8.54) | 0.602 |
| Neonatal Death | 9 (0.06) | 19 (0.13) | 2.30 (1.04–5.08) | 0.040 | 1.21 (0.98–1.51) | 1.41 (0.82–2.43) | 0.77 (0.32–1.85) | 0.554 |
| Therapeutic Hypothermia | 19 (0.12) | 23 (0.16) | 1.32 (0.72–2.42) | 0.375 | 1.03 (0.87–1.22) | 1.52 (1.01–2.28) | 0.45 (0.22–0.93) | 0.030 |
| NICU admission within 3 hours | 75 (0.48) | 97 (0.67) | 1.41 (1.04–1.91) | 0.026 | 1.08 (0.99–1.18) | 1.38 (1.12–1.69) | 0.63 (0.44–0.89) | 0.009 |

*23 cases excluded where gestation and/or mode of birth not known (20 before and 3 afterwards).

therapeutic hypothermia was reversed after 24/7 (OR 0.39; CI 0.15 to 1.00; P = 0.051) though the numbers were small. There was an overall upward trend in emergency caesarean sections in daytime births (daytime OR 1.06 pa; CI 1.03 to 1.08) unaffected by the introduction of 24/7 consultant presence. By contrast there was a rise in the trend of night-time emergency caesarean sections after July 2014 (night-time OR 1.11 pa; CI 1.01–1.22).

Other subgroup trends were broadly similar to those observed in the overall outcomes analysis described above.

**Table 4. Neonatal outcomes stratified by time of birth.**

| | Comparison between outcomes before and after July 2014 | | | | Time Series Analysis | | | |
| --- | --- | --- | --- | --- | --- | --- | --- | --- |
| | – i.e. without and with 24/7 consultant cover | | | | Overall slope 2011–2016 | Slope until July 2014 | Change in slope from July 2014 | |
| | N (%) | N (%) | Odds-Ratio | P-value | Odds-Ratio (p.a.) | Odds-Ratio (p.a.) | Odds-Ratio (p.a.) | P-value |
| | Before July 2014 | From July 2014 | (95% CI) | | (95% CI) | (95% CI) | (95% CI) | |
| **Day-time Births (08:00–19:59):** | | | | | | | | |
| Numbers of Babies | 8048 | 7414 | | | | | | |
| Primary (Composite) Outcome | 194 (2.41) | 238 (3.21) | 1.34 (1.11–1.63) | 0.003 | 1.11 (1.05–1.17) | 1.16 (1.02–1.32) | 0.91 (0.73–1.14) | 0.416 |
| Intrapartum Still Birth | 11 (0.14) | 3 (0.04) | 0.30 (0.08–1.06) | 0.062 | 0.73 (0.53–1.02) | 1.00 (0.56–1.78) | 0.39 (0.08–1.82) | 0.229 |
| Neonatal Death | 30 (0.37) | 33 (0.45) | 1.19 (0.73–1.96) | 0.481 | 1.09 (0.95–1.26) | 0.88 (0.64–1.22) | 1.53 (0.86–2.73) | 0.151 |
| Therapeutic Hypothermia | 10 (0.12) | 12 (0.16) | 1.30 (0.56–3.02) | 0.537 | 1.00 (0.79–1.27) | 1.48 (0.86–2.56) | 0.44 (0.16–1.20) | 0.108 |
| NICU admission within 3 hours | 173 (2.15) | 222 (2.99) | 1.41 (1.15–1.72) | 0.001 | 1.12 (1.05–1.18) | 1.18 (1.03–1.35) | 0.90 (0.71–1.14) | 0.376 |
| **Night-time Births (20:00–08:00):** | | | | | | | | |
| Numbers of Babies | 9276 | 8696 | | | | | | |
| Primary (Composite) Outcome | 165 (1.78) | 200 (2.30) | 1.30 (1.06–1.60) | 0.014 | 1.06 (1.00–1.13) | 1.09 (0.95–1.25) | 0.95 (0.74–1.20) | 0.644 |
| Intrapartum Still Birth | 6 (0.06) | 6 (0.07) | 1.07 (0.34–3.31) | 0.911 | 0.75 (0.53–1.07) | 0.66 (0.34–1.27) | 1.41 (0.34–5.85) | 0.635 |
| Neonatal Death | 20 (0.22) | 28 (0.32) | 1.49 (0.84–2.66) | 0.170 | 1.09 (0.93–1.29) | 0.98 (0.68–1.43) | 1.23 (0.64–2.38) | 0.540 |
| Therapeutic Hypothermia | 12 (0.13) | 13 (0.15) | 1.16 (0.53–2.53) | 0.718 | 1.05 (0.84–1.31) | 1.69 (0.97–2.93) | 0.39 (0.15–1.00) | 0.051 |
| NICU admission within 3 hours | 149 (1.61) | 183 (2.10) | 1.32 (1.06–1.64) | 0.014 | 1.07 (1.01–1.14) | 1.11 (0.96–1.28) | 0.93 (0.73–1.20) | 0.588 |

## National survey

109/196 (56%) UK maternity units responded. Of those that responded only one reported providing 24/7 labour ward consultant presence. For the 28 units delivering >5000 babies per annum, the median number of hours of consultant presence per week on labour ward was 97.5 (IQR 83–98). The number of hours of resident consultant present increased with the number of births in each unit (see Fig 2).

Perceived advantages of 24/7 consultant presence included greater stability of the service (e.g. filling middle grade doctor rota gaps, improved training opportunities, more predictable absence from daytime activities), and improved care for women and babies (e.g. greater continuity, expert review, safer care). Perceived disadvantages included higher cost and difficulty recruiting and retaining consultants to undesirable working patterns.

## Discussion

When 24/7 resident consultant presence was implemented in a large obstetric unit, the primary composite outcome was unaffected overall. Throughout the six year period there was an upward trend in the composite primary outcome, mainly accounted for by an unexpected increase in the rate of admissions to NICU within 3 hours of birth in pre-term babies. No

**Table 5. Maternal outcomes stratified by weeks gestation\*.**

| | Comparison between outcomes before and after July 2014 – i.e. without and with 24/7 consultant cover | | | | Time Series Analysis | | | |
| --- | --- | --- | --- | --- | --- | --- | --- | --- |
| | | | | | Overall slope 2011–2016 | Slope until July 2014 | Change in slope from July 2014 | |
| | N (%) | N (%) | Odds-Ratio | P-value | Odds-Ratio (p.a.) | Odds-Ratio (p.a.) | Odds-Ratio (p.a.) | P-value |
| | Before July 2014 | From July 2014 | (95% CI) | | (95% CI) | (95% CI) | (95% CI) | |
| **24 wks to 33 wks + 6 days:** | | | | | | | | |
| Women | 483 | 514 | | | | | | |
| Emergency C-Section | 261 (54.04) | 249 (48.44) | 0.80 (0.62–1.03) | 0.078 | 0.95 (0.88–1.02) | 0.86 (0.73–1.01) | 1.23 (0.92–1.64) | 0.161 |
| †Postpartum Haemorrhage | 32 (6.65) | 50 (10.62) | 1.67 (1.05–2.65) | 0.031 | 1.08 (0.95–1.23) | 1.16 (0.86–1.56) | 0.86 (0.51–1.46) | 0.583 |
| **34 wks to 36 wks + 6 days:** | | | | | | | | |
| Women | 952 | 968 | | | | | | |
| Emergency C-Section | 297 (31.20) | 340 (35.12) | 1.19 (0.99–1.44) | 0.058 | 1.05 (1.00–1.11) | 0.93 (0.83–1.05) | 1.28 (1.03–1.60) | 0.027 |
| †Postpartum Haemorrhage | 54 (5.71) | 71 (7.94) | 1.43 (0.99–2.06) | 0.060 | 1.12 (1.01–1.24) | 1.23 (0.96–1.57) | 0.84 (0.55–1.27) | 0.405 |
| **≥ 37 weeks:** | | | | | | | | |
| Women | 15676 | 14435 | | | | | | |
| Emergency C-Section | 2467 (15.74) | 2416 (16.74) | 1.08 (1.01–1.14) | 0.019 | 1.03 (1.01–1.05) | 1.02 (0.98–1.06) | 1.02 (0.95–1.10) | 0.525 |
| †Postpartum Haemorrhage | 631 (4.04) | 755 (5.63) | 1.42 (1.27–1.58) | < 0.001 | 1.10 (1.06–1.13) | 1.25 (1.16–1.34) | 0.78 (0.68–0.88) | < 0.001 |
| **Day-time Deliveries (8am to 8pm):** | | | | | | | | |
| Women | 7936 | 7320 | | | | | | |
| Emergency C-Section | 1577 (19.87) | 1619 (22.12) | 1.15 (1.06–1.24) | 0.001 | 1.06 (1.03–1.08) | 1.07 (1.01–1.12) | 0.98 (0.90–1.08) | 0.736 |
| †Postpartum Haemorrhage | 355 (4.49) | 451 (6.65) | 1.51 (1.31–1.75) | < 0.001 | 1.11 (1.07–1.16) | 1.29 (1.18–1.42) | 0.75 (0.63–0.88) | 0.001 |
| **Night-time Births (8pm to 8am):** | | | | | | | | |
| Women | 9175 | 8597 | | | | | | |
| Emergency C-Section | 1448 (15.78) | 1386 (16.12) | 1.03 (0.95–1.11) | 0.536 | 1.01 (0.98–1.03) | 0.96 (0.91–1.01) | 1.11 (1.01–1.22) | 0.025 |
| †Postpartum Haemorrhage | 362 (3.96) | 425 (5.32) | 1.36 (1.18–1.57) | < 0.001 | 1.08 (1.04–1.13) | 1.20 (1.09–1.32) | 0.82 (0.70–0.97) | 0.018 |

\*\*23 cases excluded where gestation and/or mode of birth not known (20 before and 3 afterwards).

†Blood loss data available for 17,035 (99.6%) women before July 2014, but for only 14,776 (92.8%) from July 2014.

NICU policies changed over study period that would have changed things except for the 2014 Preterm Pathway, which meant that more preterm babies were born in hospital A (rather than transferred there ex-utero from hospital B). Most babies are born at term, and represent the majority of the workload on labour wards, and therefore are the group where 24/7 consultant presence can make the greatest impact. In term babies, the upward trend in the composite outcome (specifically NICU admission and babies requiring therapeutic hypothermia) was reversed after introducing 24/7 consultant presence.

As part of the study, we conducted a similar analysis of data over the same period from the second obstetric unit (unit B) in the same NHS Trust with level 1 neonatal unit. Similar upward trends in the prevalence of emergency caesarean section (OR 1.05 pa; CI 1.03–1.07),

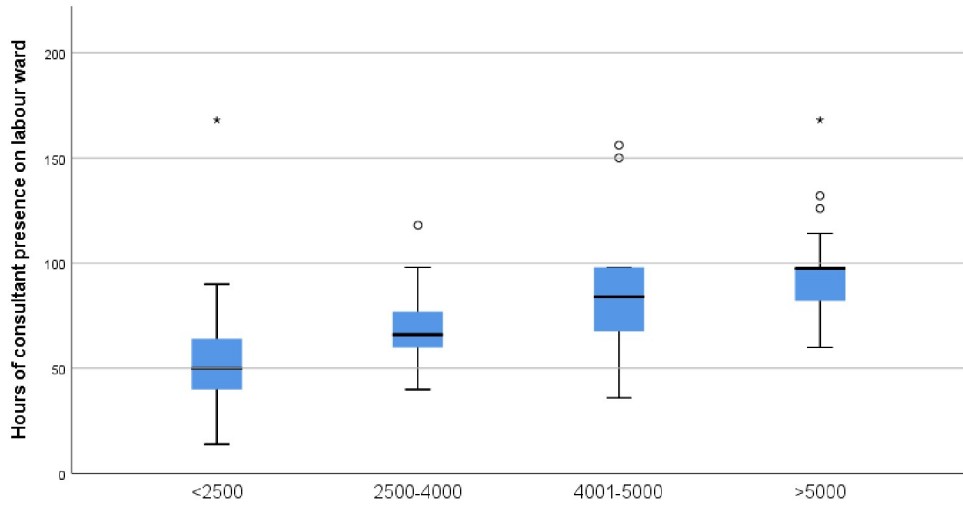

**Fig 2. Median number of hours of consultant presence by size of UK maternity units responding to survey.**

ventilation (1.10 pa; CI 0.99–1.21) and postpartum haemorrhage (OR 1.13; CI 1.08–1.18) were observed. By contrast there was little change in the rate of the composite neonatal outcome (64/9672 = 0.66% before July 2014 compared to 64/9335 = 0.69% afterwards). Comparisons between the units are inappropriate given the much lower rate of NICU admissions in unit B (a level 1 neonatal unit).

## Strengths and limitations

The study was based in a public hospital that had actually implemented 24/7 consultant presence exactly as recommended by UK RCOG. To our knowledge, this is the only unit in England to publish neonatal and maternal outcomes evaluating effects of introducing 24/7 labour ward consultant presence. The study has two major strengths. The dataset was large enough (33,434 babies and 33,051 women) to be able to include serious outcomes which could be plausibly be affected by consultant presence during labour and which reflect national priorities [24]. The ITS study design, a robust quasi-experimental approach to evaluating policy interventions, explored the specific effect of the staffing changes, accounting for underlying secular trends in outcomes which would not be detected by a simple before-and-after study [25]. The ITS approach has been recommended for use in service-delivery research as suggestive of causal relationships when appropriate controls are unavailable [26].

The primary study limitation is that it was retrospective, using routinely collected data: no formal evaluation of the policy change was planned prospectively, and no data was available to measure change in skill mix or fidelity to the policy. Some outcomes of interest were not possible to explore, as data was either not routinely gathered or was of insufficient quality due to a change in maternity information system halfway through the study, meaning that we excluded certain outcomes as they were not consistently gathered across the time period, including second stage Caesarean section, failed instrumental delivery, dual instrumentation, and maternal admission to high dependency or intensive care. It was disappointing that the data on maternal haemorrhage was not of good enough quality given that a substantial decrease was shown in this after July 2014 consistent with a beneficial impact of 24/7 presence. Since the proportion

of women for whom blood loss was recorded fell from 99.6% before 24/7 consultant presence to only 92.8% after July2014 this must be treated with caution.

The power calculation relied on the published birth rate available at the time of protocol development (6,500 per year), which was greater than the actual births occurring during the measurement period (5,775), however the results achieved statistical significance. The low frequency of individual adverse events meant that the study was not powered to detect a difference in each outcome and a composite was required, however the most important neonatal outcomes were included. The five newly recruited consultants may have differed from the existing obstetric team: it was not possible to account for any variations in the context of this pragmatic retrospective study which represented a real-life upscaling of the obstetric workforce, recruiting consultants meeting at least the minimum standard for the role in the UK.

## Implications for research and practice

Our study offers more robust insights than previously available into 24/7 cover: previous work exploring the relationship between consultant presence and maternal and neonatal outcomes was limited because no unit had implemented full 24/7 cover [13, 14], therefore could only compare parts of the week with or without consultant presence. While UK-focused, the findings are of relevance to other contexts with similar models of maternity care where senior obstetricians are not usually present 24/7.

The potential neonatal benefit of 24/7 labour ward consultant presence in babies ≥37 weeks is clinically plausible, as obstetric interventions are most likely to benefit term babies, who constitute >90% of births. The time of day analysis supports this, as after consultants became resident at night there was a rise in the trend for emergency caesarean sections at night and simultaneously the trend in babies born at night who required therapeutic hypothermia reduced. The overall numbers of babies who may have benefitted from this policy in the three years included is relatively small; however, not only is each case of admission to NICU or therapeutic cooling of great significance to families involved, it is extremely costly for the health service, in both direct care and potential litigation.

The reversal of the upward trend shown in maternal haemorrhage would be of major importance as this is a frequent direct causes of maternal death [27] however these results must be treated with caution due to a reduction in documentation after introduction of a new maternity information system.

Differences in qualitative outcomes and staff satisfaction in this unit were not included in this study, as they were evaluated previously: a survey of maternity care staff in our study site was undertaken by a different group of researchers, one year after the introduction of the 24/7 consultant presence [28]. The survey demonstrated wide-spread satisfaction with the change, and a strong perception of improvements in safety, efficiency, teaching opportunities and morale. Other qualitative research has reported on the positive effects consultant presence can have on teamwork, training, and predictable work patterns [29]. Clinical directors in our national survey agreed that these qualitative outcomes were among the most important potential benefits of increasing consultant presence. Given the high prevalence of burnout reported among obstetricians, especially trainees [30], interventions to improve staff morale and safety are critical. Drawbacks to 24/7 consultant presence included the high cost of additional consultants, dissatisfaction among some at working resident nights and possible diversion of consultant time from day-time work (both clinical sessions and availability for other roles in management, teaching or research).

Recently the UK RCOG changed their recommendations, suggesting that alternatives to 24/7 consultant presence should be explored [31, 32], which was beyond the scope of our

study. Units that wish to consider expanding the number of resident consultant hours on labour ward should clearly identify what benefits they would expect to see, consider comparisons with alternatives (e.g. increasing non-consultant-grade obstetrician or midwife staffing), and prospectively collect pre- and post-intervention data to evaluate impact. The potential benefit to term babies and staff wellbeing must be balanced against the cost of employing consultants, the need for sufficient day-time cover and the effect of shift patterns on recruiting and retaining consultants.

## Conclusion

There was an overall upward trend in the primary composite neonatal outcome over the study period which was not affected by 24/7 consultant presence on the labour ward; however, it did reverse the trend in babies born at $\geq$37 weeks gestation. Findings suggest that 24/7 consultant presence may reverse upward trends in a major cause of maternal death, post-partum haemorrhage, though data on this was incomplete. This is the first study to suggest that a policy of 24/7 consultant presence on labour ward appears to be of benefit to term babies who constitute the majority of the workload and whose outcomes are most likely to be impacted by obstetric interventions.

## Supporting information

**S1 File. Survey of clinical directors.**
(PDF)

## Acknowledgments

Clinical Advisory Group (University Hospitals Birmingham): Irshad Ahmed, Carla Jones-Charles, Katherine Barber, Mike Wyldes, Donna Prestage.

Analysis of survey results: Ellie Jones (Institute of Applied Health Research, University of Birmingham, UK).

## Author Contributions

**Conceptualization:** Sharon Morad, David Pitches, Christine MacArthur, Sara Kenyon.

**Data curation:** Sharon Morad, David Pitches, Alan Girling, Vikki Fradd.

**Formal analysis:** Sharon Morad, David Pitches, Alan Girling.

**Investigation:** Sharon Morad.

**Methodology:** Sharon Morad, David Pitches, Alan Girling, Beck Taylor, Christine MacArthur, Sara Kenyon.

**Project administration:** Sharon Morad, David Pitches.

**Software:** Alan Girling.

**Supervision:** Christine MacArthur, Sara Kenyon.

**Validation:** Alan Girling, Vikki Fradd.

**Writing – original draft:** Sharon Morad, David Pitches.

**Writing – review & editing:** Sharon Morad, David Pitches, Alan Girling, Beck Taylor, Vikki Fradd, Christine MacArthur, Sara Kenyon.

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
