## [Decision Letter · Decision Letter 0]

23 Dec 2020

PONE-D-20-30095

24 hour consultant obstetrician presence on the labour ward and intrapartum outcomes in a large unit in England: a time series analysis.

PLOS ONE

Dear Dr. Taylor,

Thank you for submitting your manuscript to PLOS ONE. After careful consideration, we feel that it has merit but does not fully meet PLOS ONE’s publication criteria as it currently stands. Therefore, we invite you to submit a revised version of the manuscript that addresses the points raised during the review process.

We look forward to receiving your revised manuscript.

Kind regards,

Andrew Sharp, PhD

Academic Editor

PLOS ONE

Journal Requirements:

2. For more information on PLOS ONE's expectations for statistical reporting, please see https://journals.plos.org/plosone/s/submission-guidelines.#loc-statistical-reporting. Please update your Methods and Results sections accordingly.

3.Thank you for stating the following in the Competing Interests section:

"Sharon Morad reports that she is married to one of the co-authors (DP).

David Pitches reports that he is married to the corresponding author (SM).

Alan Girling has nothing to disclose.

Vikki Fradd has nothing to disclose.

Beck Taylor reports funding from the NIHR during the study duration.

Christine MacArthur reports funding from the NIHR during the study duration.

Sara Kenyon reports funding from the NIHR during the study duration."

4.We note that you have indicated that data from this study are available upon request. PLOS only allows data to be available upon request if there are legal or ethical restrictions on sharing data publicly. For information on unacceptable data access restrictions, please see http://journals.plos.org/plosone/s/data-availability#loc-unacceptable-data-access-restrictions.

Additional Editor Comments (if provided):

please make it clear in the results section of the abstract what the primary outcome increase actually is

Reviewers' comments:

Reviewer's Responses to Questions

**Comments to the Author**

1. Is the manuscript technically sound, and do the data support the conclusions?

Reviewer #1: Yes

2. Has the statistical analysis been performed appropriately and rigorously? 

Reviewer #1: Yes

3. Have the authors made all data underlying the findings in their manuscript fully available?

Reviewer #1: No

4. Is the manuscript presented in an intelligible fashion and written in standard English?

Reviewer #1: Yes

5. Review Comments to the Author

Reviewer #1: The authors seek to evaluate the impact of 24/7 consultant presence in one maternity unit in England on maternal and perinatal outcomes, using a retrospective data analysis of routinely collected hospital data, using a time series analysis. This is an original study, on a topic that is highly relevant for practising obstetricians and health managers and the limitations of a retrospective analysis of routinely collected data are highlighted.

Major issues

This paper focuses on neonatal outcomes and more information on maternal outcomes would be useful. It is such a shame that the maternal data was not of high enough quality. It is clear in the paper there is no information on certain parameters for ITU admission etc, but I wonder if any of the other relevant items from the COS/systematic review such as maternal deaths, start of labour, tears, lengths of waiting time, lengths of second stage or other outcomes are available.

If not could a few more sentences be added explaining the maternal outcomes a little further, so the paper is covering maternal, as well as perinatal outcomes.

Devane D, Begley CM, Clarke M, Horey D, OBoyle C. Evaluating maternity care: a core set of outcome measures. Birth. 2007 Jun;34(2):164-72. doi: 10.1111/j.1523-536X.2006.00145.x. PMID: 17542821.

195 – Is the survey results already published elsewhere, or is this paper the primary dissemination of these results. If so more detail is needed on the methods of the survey. The survey aspect is not reproducible in future studies, with the current description of methods.

Minor issues

Abstract

The wording of the abstract results could be altered slightly. On my initial (quick read) I saw that the primary outcome had significantly increased, and this was not the conclusion. It is clear throughout the paper, that this is expected. However, slightly adjusting the wording in the abstract (which is all many readers will read) could make the message from the paper a little clearer in the abstract.

Should there be a short comment on maternal outcomes in the conclusion?

Line 55 -Add ref for the confidential enquiries

Line 120 – why was the 24/7 consultant cover ceased?

Table 1 – please write out IMD quartile in full, rather than abbreviation

Table 2 – stillbirth (ante or post partum) – should this be intra partum, rather than post?

242 – Could the maternal and neonatal outcomes be described in two separate paragraphs, rather than the PPH sentence mixed with information about the neonates

6. PLOS authors have the option to publish the peer review history of their article (what does this mean?). If published, this will include your full peer review and any attached files.

Reviewer #1: No

---

## [Author Response · Author response to Decision Letter 0]

26 Feb 2021

We thank the Editor and the Reviewer for your comments. We have addressed each recommendation systematically, explained in the tables at the end of our covering letter, and highlighted in the attached manuscript. Our response is copied again below but may be easier to read in the tabular format in attached documents.

• Competing interests [original statement plus recommended additional text] Sharon Morad reports that she is married to one of the co-authors (DP). David Pitches reports that he is married to the corresponding author (SM). Alan Girling has nothing to disclose. Vikki Fradd has nothing to disclose. Beck Taylor reports funding from the NIHR during the study duration. Christine MacArthur reports funding from the NIHR during the study duration. Sara Kenyon reports funding from the NIHR during the study duration. Our competing interests statement does not alter our adherence to PLOS ONE policies on sharing data and materials.

• Restrictions on sharing of data and materials There are legal and ethical restrictions on sharing the data set. The data is routine English National Health Service patient data, and contains potentially identifiable or sensitive patient information. A data sharing agreement is in place between the NHS trust and the University of Birmingham to enable analysis of data, but does not permit sharing beyond the research team. Ethical approval provided by the University of Birmingham Research Ethics Committee, reference ERN_18-0660, contact s.l.cottam@bham.ac.uk. The ethical approval was requested for access to data by the research team only, and not by any external parties.

Editor’s comments 

Heading font size amended to fit style requirements

Paragraph single spaced in error amended to double space

Manuscript and title page separated into different files

File names of figures amended (we could not identify specific requirements for manuscript/title file naming)

References added and formatted

Author contributions have not been changed as the guidance states “If you would like the equal contributions notes to read differently, please specify in your manuscript (e.g., "AR and MM are Joint Senior Authors").” Please let us know if you require a different format.

2. For more information on PLOS ONE's expectations for statistical reporting, please see https://journals.plos.org/plosone/s/submission-guidelines.#loc-statistical-reporting. Please update your Methods and Results sections accordingly. 

We have added one sentence to the methods about Z-tests and significance thresholds at the end of p10. "P-values are derived from two-tailed Z-tests with P = 0.05 as the threshold of significance throughout."

Numerators and denominators added to the percentages in the abstract for the primary outcome.

p-values have been rounded to 3 decimal places where they were 4 in original submission. 

3.Please confirm that the competing interests statement does not alter your adherence to all PLOS ONE policies on sharing data and materials, by including the following statement: "This does not alter our adherence to PLOS ONE policies on sharing data and materials.” (as detailed online in our guide for authors http://journals.plos.org/plosone/s/competing-interests). If there are restrictions on sharing of data and/or materials, please state these. Please include your updated Competing Interests statement in your cover letter; we will change the online submission form on your behalf 

Added to the cover letter: 

Statement that competing interests statement does not alter our adherence to PLOS ONE policies on sharing data and materials to competing interests statement 

Statement of restrictions on sharing of data and/or materials

4.We note that you have indicated that data from this study are available upon request. PLOS only allows data to be available upon request if there are legal or ethical restrictions on sharing data publicly. For information on unacceptable data access restrictions, please see http://journals.plos.org/plosone/s/data-availability#loc-unacceptable-data-access-restrictions. In your revised cover letter, please address the following prompts:

We will update your Data Availability statement on your behalf to reflect the information you provide. There are legal and ethical restrictions on sharing the data set. 

The data is routine English National Health Service patient data, and contains potentially identifiable or sensitive patient information. 

A data sharing agreement is in place between the NHS trust and the University of Birmingham to enable analysis of data, but does not permit sharing beyond the research team. 

Ethical approval provided by the University of Birmingham Research Ethics Committee, reference ERN_18-0660, contact s.l.cottam@bham.ac.uk. The ethical approval was requested for access to data by the research team only, and not by any external parties.

5. Please make it clear in the results section of the abstract what the primary outcome increase actually is 

Abstract amended to state “The prevalence of the primary outcome increased by 0.65%”

Reviewers' comments: 

1. Is the manuscript technically sound, and do the data support the conclusions?

Reviewer #1: Yes No action

2. Has the statistical analysis been performed appropriately and rigorously? 

Reviewer #1: Yes No action

3. Have the authors made all data underlying the findings in their manuscript fully available?

Reviewer #1: No Please see earlier amendments regarding data availability

4. Is the manuscript presented in an intelligible fashion and written in standard English?

Reviewer #1: Yes No action

Major issues

This paper focuses on neonatal outcomes and more information on maternal outcomes would be useful. It is such a shame that the maternal data was not of high enough quality. It is clear in the paper there is no information on certain parameters for ITU admission etc, but I wonder if any of the other relevant items from the COS/systematic review such as maternal deaths, start of labour, tears, lengths of waiting time, lengths of second stage or other outcomes are available. If not could a few more sentences be added explaining the maternal outcomes a little further, so the paper is covering maternal, as well as perinatal outcomes.

Devane D, Begley CM, Clarke M, Horey D, OBoyle C. Evaluating maternity care: a core set of outcome measures. Birth. 2007 Jun;34(2):164-72. doi: 10.1111/j.1523-536X.2006.00145.x. PMID: 17542821.

We agree that it is disappointing that the maternal data was not of sufficient quality. 

We have separated maternal and neonatal secondary outcome findings under new subheadings. 

We have added further discussion of maternal outcomes and rationale for inclusion, along with more detail regarding the data limitations in the discussion section.

195 – Is the survey results already published elsewhere, or is this paper the primary dissemination of these results. If so more detail is needed on the methods of the survey. The survey aspect is not reproducible in future studies, with the current description of methods.

The survey has not been published previously. While not the primary focus of the paper, we welcome the suggestion to strengthen the description of methods. We have done so, and have included the survey questions as a supplementary file. Questionnaire sent as supplementary information.

Minor issues

Abstract

The wording of the abstract results could be altered slightly. On my initial (quick read) I saw that the primary outcome had significantly increased, and this was not the conclusion. It is clear throughout the paper, that this is expected. However, slightly adjusting the wording in the abstract (which is all many readers will read) could make the message from the paper a little clearer in the abstract.

Conclusion in abstract amended to better reflect the findings

Should there be a short comment on maternal outcomes in the conclusion?

Comment added: Findings suggest that 24/7 consultant presence may reverse upward trends in a major cause of maternal death, post-partum haemorrhage, though data on this was incomplete.

Line 55 -Add ref for the confidential enquiries

Reference added

Line 120 – why was the 24/7 consultant cover ceased? 

The 24/7 consultant cover ceased due to funding constraints (added to text).

Table 1 – please write out IMD quartile in full, rather than abbreviation 

IMD amended and written in full

Table 2 – stillbirth (ante or post partum) – should this be intra partum, rather than post? 

Amended to intrapartum

242 – Could the maternal and neonatal outcomes be described in two separate paragraphs, rather than the PPH sentence mixed with information about the neonates 

Maternal and neonatal secondary outcomes separated under new subheadings

---

## [Editor Report · Decision Letter 1]

15 Mar 2021

24 hour consultant obstetrician presence on the labour ward and intrapartum outcomes in a large unit in England: a time series analysis.

PONE-D-20-30095R1

Dear Dr. Taylor,

We’re pleased to inform you that your manuscript has been judged scientifically suitable for publication and will be formally accepted for publication once it meets all outstanding technical requirements.

Kind regards,

Andrew Sharp, PhD

Academic Editor

PLOS ONE
---

## [Editor Report · Acceptance letter]

19 Mar 2021

PONE-D-20-30095R1 

24 hour consultant obstetrician presence on the labour ward and intrapartum outcomes in a large unit in England: a time series analysis 

Dear Dr. Taylor:

I'm pleased to inform you that your manuscript has been deemed suitable for publication in PLOS ONE. Congratulations! Your manuscript is now with our production department. 

Kind regards, 

on behalf of

Dr. Andrew Sharp 

Academic Editor

PLOS ONE